# Adipokine levels and their association with clinical disease severity in patients with dengue

Heshan Kuruppu[1], W. P Rivindu H. Wickramanayake[1], Chandima Jeewandara[1], Deneshan Peranantharajah[1], H. S. Colambage[1], Lahiru Perera[1], Laksiri Gomes[1], Ananda Wijewickrama[2], Graham S. Ogg[3], Gathsaurie Neelika Malavige [1,3]*

1 Allergy, Immunology and Cell Biology Unit, Department of Immunology and Molecular Medicine, University of Sri Jayewardenepura, Nugegoda, Sri Lanka, 2 National Institute of Infectious Diseases, Angoda, Sri Lanka, 3 MRC Human Immunology Unit, MRC Weatherall Institute of Molecular Medicine, University of Oxford, Oxford, United Kingdom

* gathsaurie.malavige@ndm.ox.ac.uk

**Data Availability Statement:** Data is available within the manuscript and the supporting files.

**Funding:** We are grateful to the NIH, USA (grant number 5U01AI151788-02 to GNM) and the UK

## Abstract

Adipokines have not been studied in acute dengue, despite their emerging role in inducing and regulating inflammation. Therefore, we sought to identify adipokine levels in patients with varying severities of acute dengue to understand their role in disease pathogenesis. We determined the levels of leptin, resistin, omentin, adiponectin, as well as IFNβ, and NS1 using quantitative ELISA in patients with dengue fever (DF = 49) and dengue haemorrhagic fever (DHF = 22) at admission (febrile phase) and at the time of discharge (recovery phase). The viral loads and serotypes of all samples were quantified using quantitative real-time RT-PCR. Resistin levels (p = 0.04) and omentin (p = 0.006) levels were significantly higher in patients who developed DHF. Omentin levels in the febrile phase also correlated with the AST (Spearman's r = 0.38, p = 0.001) and ALT levels (Spearman's r = 0.24, p = 0.04); as well as serum leptin levels with both AST (Spearman's r = 0.27, p = 0.02) and ALT (Spearman's r = 0.28, p = 0.02). Serum adiponectin levels in the febrile phase did not correlate with any of the other adipokines or with liver enzymes, but inversely correlated with CRP levels (Spearman's r = -0.31, p = 0.008). Although not significant (p = 0.14) serum IFNβ levels were lower in the febrile phase in those who progressed to develop DHF (median 0, IQR 0 to 39.4 pg/ml), compared to those who had DF (median 37.1, IQR 0 to 65.6 pg.ml). The data suggest that adipokines are likely to play a role in the pathogenesis of dengue, which should be further explored for the potential to be used as prognostic markers and as therapeutic targets.

## Author summary

Adipokines have not been studied in acute dengue, despite their emerging role in inducing and regulating inflammation. Therefore, we sought to identify adipokine levels in patients with varying severities of acute dengue to understand their role in disease

Medical Research Council (to GSO). The funders had no role in study design, data collection and analysis, decision to publish, or preparation of the manuscript.

**Competing interests:** The authors have no competing interests.

pathogenesis. We determined the levels of leptin, resistin, omentin, adiponectin, IFNβ, and NS1 in patients with dengue fever (DF) and dengue haemorrhagic fever (DHF) at admission (febrile phase) and at the critical phase. Resistin levels and omentin levels were significantly higher in patients who developed DHF. Omentin levels in the febrile phase also correlated with the AST and ALT levels; as well as serum leptin levels with both AST and ALT. Serum adiponectin levels in the febrile phase did not correlate with any of the other adipokines or with liver enzymes, but inversely correlated with CRP levels. The data suggest that adipokines are likely to play a role in the pathogenesis of dengue, which should be further explored for the potential to be used as prognostic markers and as therapeutic targets.

## Introduction

Dengue is one of the most important and rapidly emerging vector-borne diseases affecting over 195 countries globally [1]. The age stratified incidence, the disability adjusted life years and deaths due to dengue have gradually increased over the last 30 years [1]. Due to climate change and rapid urbanization, it is predicted that the global incidence will further increase with individuals being infected with multiple dengue virus (DENV) serotypes [2]. It is estimated that 390 million individuals are infected by the virus annually, with 25% of these infections resulting in symptomatic dengue infections [3]. Although the majority of those who are infected with the DENV develop asymptomatic or mild illness, a proportion of infected individuals develop complications in the form of dengue haemorrhagic fever (DHF), including shock, severe bleeding and organ dysfunction [4]. Those with diabetes, obesity, hypertension and asthma are more likely to develop severe dengue and complications [5–8].

Endothelial dysfunction leading to vascular leak is one of the most important factors leading to severe dengue [9]. Although the majority of those infected with DENV recover without development of endothelial dysfunction, Individuals who develop DHF exhibit an immune response that is altered and dysfunctional with high levels of inflammatory cytokines, chemokines and inflammatory lipid mediators along with high levels of immunosuppressive cytokines such as IL-10[10–12]. An altered and dysfunctional immune response is also seen in those who develop severe COVID-19, influenza, and many other infectious diseases [13,14]. The presence of metabolic diseases such as obesity, diabetes and hypertension are risk factors for development of severe COVID-19, influenza and dengue [15,16]. These metabolic diseases are associated with altered and dysfunctional immune responses, which may impair antiviral immunity and promote a broad proinflammatory response [13,14]. In both influenza and SARS-CoV-2 infection, antiviral responses have shown to be impaired and delayed in obese individuals [17] and we have shown that those who have severe dengue too have an impaired type I IFN response [18]. The ongoing low-grade inflammation seen in obesity and diabetes facilitates a broad proinflammatory immune response, rather than a robust antiviral response [17,19]. Although both diabetes and obesity are important risk factors for development of severe dengue, the mechanisms for severe dengue in those with metabolic disease have not been studied.

Adipose tissue is an important source of adipokines, such as leptin, adiponectin and cytokines such as IL-6, MCP-1 and TNFα, which play an important role in regulation of the immune system [20]. While some of the adipokines such as TNFα, IL-6 and resistin are potent mediators of inflammation, adipokines such as adiponectin and omentin are known to have anti-inflammatory roles [21]. Patients who had severe COVID-19, were shown to have higher

levels of resistin, IL-6, TNFα and reduced adiponectin/leptin ratios suggesting that adipokines are likely to play an important role in disease pathogenesis [22]. Serum adiponectin levels were also shown to correlate with SARS-CoV-2 receptor binding domain specific antibodies in previously infected individuals who received the BNT162b2 vaccine [20]. In contrast, although adiponectin is known as an anti-inflammatory adipokine, higher levels in influenza infection in elderly individuals were associated with more severe lung pathology and more severe outcomes [23]. Therefore, it appears that different adipokines could be playing multiple and varied roles in different virus infections.

Although both obesity and diabetes are known risk factors for development of severe dengue, their role in dengue has not been previously investigated. Furthermore, as the type I interferon response was shown to be impaired in patients with severe dengue, it would be important to investigate the relationship between a defective type I interferon response, adipokines and clinical disease severity. Currently there are no antivirals or drugs for treatment of dengue, and all dengue patients are monitored for development of complications so that timely fluid management can be initiated. In order to further understand the pathogenesis of dengue and to develop therapeutics for treatment of dengue, it would be important to understand the role of adipokines in dengue and their association with clinical disease severity, viraemia and the antiviral responses. In this study, we have measured the levels of different adipokines, interferon β, viral loads, dengue NS1 levels in patients with varying severity of dengue, during the febrile phase and at the time of discharge from hospital, to better understand the role of adipokines and the type I interferon responses in dengue.

## Methods

### Ethics statement

Ethical approval was obtained from the Ethics Review Committee of the Faculty of Medical Sciences, University of Sri Jayewardenepura (ethics application number:58/19). All participants gave informed written consent.

### Patients with acute dengue

Adult patients with clinical features suggestive of an acute dengue infection were recruited from the National Institute of Infectious Disease, Sri Lanka following informed written consent from November 2021 to August 2022. In this prospective study, 140 patients were screened and 71 were included in the study. All those with clinical evidence of other acute infections (urinary tract infection, cellulitis, or tonsillitis), chronic diseases such as chronic kidney or liver disease were excluded from the study. Blood samples were obtained from these 71 patients, in the febrile phase ($\leq$4 days since onset of illness) named as A sample, and from hospital (days 7 to 8 since onset of illness, named B sample). None of the patients had any evidence of plasma leakage or any criteria that would indicate the diagnosis of DHF at the time of recruitment. The day on which the patient first developed fever was considered as the first day of illness. All clinical symptoms and laboratory results were recorded several times of the day.

Evidence of fluid leakage was assessed using an ultrasound scan to detect pleural effusions and ascites. Disease severity was classified according to the 2011 World Health Organization (WHO) dengue diagnostic criteria for dengue [4]. Accordingly, patients with a rise in haematocrit >20% of the baseline, or those who had ultrasound scan evidence of plasma leakage were classified as having DHF. Shock was defined as the presence of a narrowing pulse pressure of 20 mm Hg in patients with DHF. According to the above criteria, 22 were classified as having DHF, as they subsequently progressed to developed criteria to be classified as DHF, during the course of their illness, and 49 patients as DF. In order to measure adipokines and

IFN-beta levels in a healthy state, 18 healthy individuals, who were matched for age and gender were recruited. 9/71 (12.6%) of those with acute dengue and 2/18 (11.1%) of healthy individuals had metabolic disease.

### Serotyping of DENV and assessment of viral copy numbers

The presence of an acute dengue infection was confirmed in all patients using quantitative real-time PCR, and DENV serotypes and viral copy numbers were quantified. Viral RNA in serum was extracted using a QIAamp Viral RNA Mini Kit (Qiagen, USA, cat:52906) and the viral copy numbers were quantified as previously described [24]. Briefly, a multiplex quantitative real-time PCR was performed using the CDC real-time PCR assay and was modified to quantify DENV. Oligonucleotide primers and a dual-labelled probe for DENV serotypes 1, 2, 3, and 4 (Life Technologies, India) were used based on published sequences.

### Quantification of adipokine levels and IFNβ levels in patients with acute dengue

Levels of leptin, resistin, adiponectin, and omentin were measured in serum stored at -80˚C using quantitative ELISA assays (Abcam, UK) according to the manufacturer's instructions. Assays were performed in serum diluted at given ratios according to the protocol for the assessment of adipokine levels. Assays were performed according to the manufacturer's instructions. A pre-coated IFNβ quantitative ELISA (R&D systems, DuoSet, USA) was used to measure cytokine levels, according to the manufacturer's instructions. Adipokines and IFNβ levels were measured in serum samples obtained in the febrile phase and the samples taken at the time of discharge, and in healthy individuals. These assays were interpreted using a four-parameter logistic graph as per the protocol.

### Quantification of dengue NS1 levels in patients with acute dengue

A commercial assay was used to semi-quantitatively measure NS1 antigen levels in patient sera (PLATELIA DENGUE NS1 Ag). The results were interpreted as ratios according to the manufacturer's instructions.

### Statistical analysis

Statistical analysis was done using GraphPad Prism version 9.5.1 (Dotmatics, California, USA). As the data were not normally distributed, differences in adipokines for different clinical disease severity were compared using the Mann-Whitney U test (two tailed). Spearman rank order correlation coefficient was used to evaluate the correlation between variables including the association between adipokines and laboratory parameters. To address the potential heterogeneity of variances in our data, we conducted a test for equal variances using Levine's test based on absolute deviation from the mean. The results of this test indicated that there were no significant differences in the variances among the samples (p value > 0.05).

## Results

### Clinical and laboratory characteristics of the patients

The clinical and laboratory features of the patients with DF and those who progressed to develop DHF are shown in Table 1. There were similar proportions of males in those with DF 25/49 (51.02%) and DHF 12/22 (57.14%) and the mean ages of those with DF was 31.53 years (SD± 15.20) and for DHF was 30.64 years (SD± 11.54). None of the patients with DHF

**Table 1. Clinical and laboratory characteristics of patients with DF and DHF.**

| Clinical and laboratory features | DF (n = 49) N (%) | DHF (n = 22) N (%) |
|---|---|---|
| Age | 26, 20 to 40.5 | 27, 21 to 39 |
| Gender | Male– 52% Female– 48.97% | Male– 59% Female– 40.90% |
| Fever | 49(100%) | 22(100%) |
| Headache | 38(77.5) | 19(86.36) |
| Myalgia | 30(61.22) | 22(100%) |
| Arthralgia | 24(48.9%) | 20(90.9%) |
| Nausea | 20(40.8%) | 21(95.4%) |
| Anorexia | 18(36.7%) | 20(90.9%) |
| Vomiting | 15(30.6%) | 19(86.3) |
| Pleural effusions | 0 | 0 |
| Ascites | 0 | 22(100%) |
| Bleeding | 0 | 0 |
| Shock | 0 | 0 |
| Platelet counts | | |
| <50,000 | 13(26.5%) | 19(86.3%) |
| <20,000 | 4(8.1%) | 5(22.7%) |

developed shock or bleeding manifestations. The laboratory parameters with adipokine levels of these patients are shown in Table 2.

## Adipokine levels in patients with acute dengue

Resistin, omentin, leptin, and adiponectin levels were measured in all 71 patients in samples obtained during the febrile (sample A) and at the time of discharge (sample D). Resistin levels were significantly higher in patients who proceeded to develop DHF (p = 0.04) compared to those who had DF (Fig 1A). Serum omentin levels were also significantly higher (p = 0.006) in

**Table 2. Adipokine and other laboratory parameters in the febrile and recovery phases in patients with acute dengue.**

| Parameter | DF N = 49 (Median, Interquartile range | | DHF N = 22 (Median, Interquartile range) | |
|---|---|---|---|---|
| | Febrile phase | Recovery phase/ at the time of discharge | Febrile phase | Recovery phase/at the time of discharge |
| | (A) | (D) | (A) | (D) |
| Resistin (ng/ml) | 10.9, 7.1 to 14.6 | 9.5, 5.8 to 15 | 14.2, 9.2 to 20.7 | 11.3, 7 to 14 |
| Omentin (ng/ml) | 565.8, 370 to 863.2 | 451.7, 286.7 to 714.7 | 974.5, 504.5 to 1091 | 743.6, 350.3 to 1105 |
| Leptin (ng/ml) | 10.4, 4.8 to 15 | 9.3, 5.6 to 13.2 | 7.03, 2.6 to 16.9 | 9.8, 4.5 to 15.9 |
| Adiponectin (μg/ml) | 48, 35.4 to 59.2 | 47, 33.1 to 61.9 | 54.9, 39.2 to 76.5 | 48.6, 32 to 76.5 |
| Viral loads (copies/ml) | 10.5, 0.45 to 344.9 | 0, 0 to 0.75 | 26, 0 to 103.9 | 1.2, 0 to 5.2 |
| NS1 antigen levels | 3.2, 0.19 to 4.2 | 0.34, 0.07 to 4.0 | 2.9, 0.06 to 4 | 0.3, 0.17 to 1 |
| IFNβ (pg/ml) | 20.5, 0 to 63.5 | 40.2, 0 to 72.4 | 0, 0 to 38.5 | 38.9, 0 to 102.3 |
| Platelet counts ($10^3$ / uL) | 153, 136 to 180 | 100, 68 to 121 | 129, 93.2 to 153.5 | 40.5, 20.5 to 54 |
| Leucocytes ($10^3$ / uL) | 4.1, 2.8 to 4.8 | 3.8, 2.6 to 4.6 | 3.3, 2.8 to 4.4 | 3.8, 2.8 to 6.4 |
| Neutrophils (%) | 65.9, 61.5 to 74 | 45.3, 34.6 to 57.9 | 65.5, 59.5 to 79.9 | 44.9, 37.8 to 59.4 |
| Lymphocytes (%) | 25, 17.5 to 31 | 40.9, 30.2 to 51.6 | 19.6, 13.8 to 33 | 38.4, 26.9 to 46.9 |
| CRP (mg/L) | 12.4, 9.9 to 19.5 | Not performed | 16.5, 9.2 to 24.9 | Not performed |
| AST (U/L) | 46.5, 34.3 to 86.3 | Not performed | 73.5, 43 to 127 | Not performed |
| ALT (U/L) | 48, 27 to 103.5 | Not performed | 53, 30.5 to 107 | Not performed |
| PCV (%) | 39, 35.5 to 41 | Not performed | 39, 36.7 to 41.2 | Not performed |

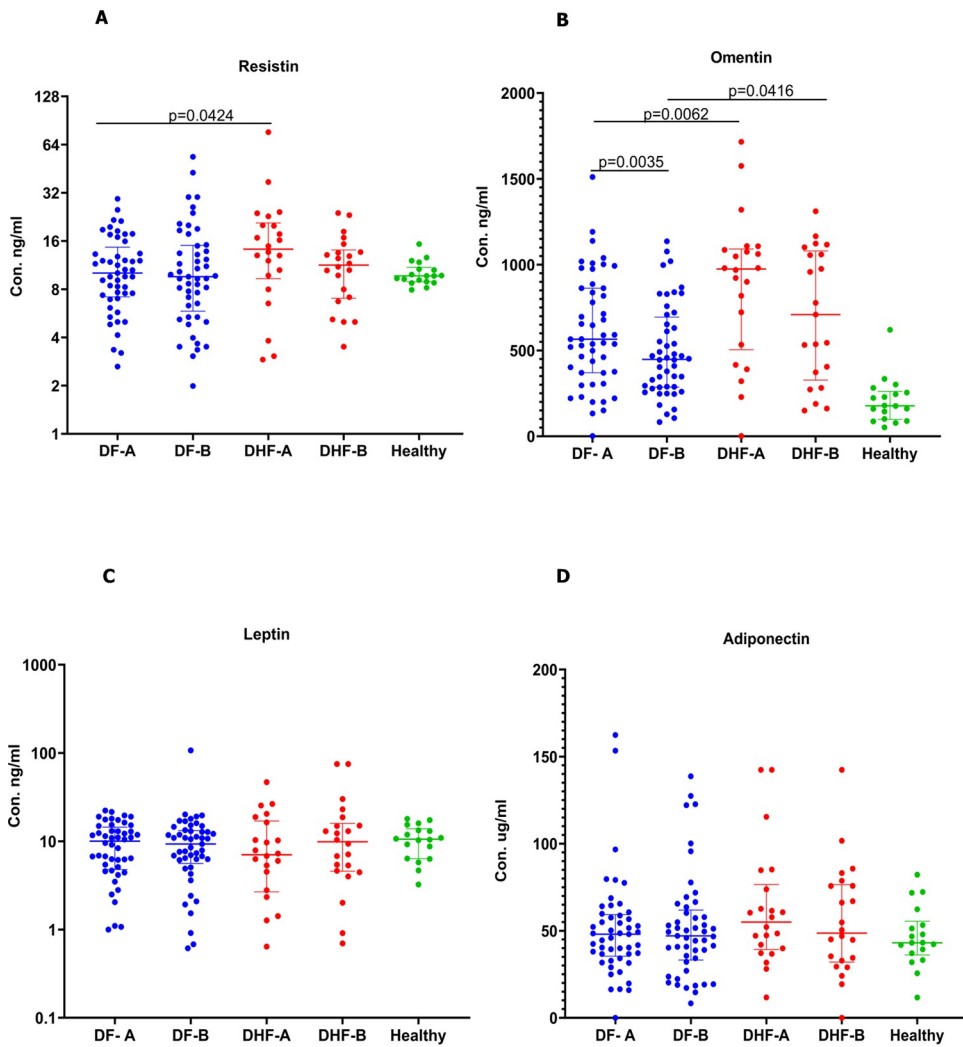

**Fig 1. Adipokine levels in patients with acute dengue.** Resistin (A), omentin (B), leptin (C), and adiponectin (D) levels were measured in patients with DF (n = 49) and DHF (n = 22) during the febrile and recovery phases, using a quantitative ELISA. The Wilcoxon matched pair singed-rank test was used to compare the levels of adipokines of the febrile (A sample) during the febrile phase (day 3 and 4 since onset of illness) and on day 5 to 6 since onset of illness (B sample) of patients with DF and DHF. The Mann-Whitney U test (two tailed) was used to calculate the differences in the means in the DF and DHF cohorts. The error bars indicate the median and the interquartile ranges.

those who progressed to develop DHF compared to those with DF (Fig 1B). The levels were still significantly higher at the time of discharge (p = 0.03), in patients with DHF compared to DF. The omentin levels declined from the levels observed in the febrile phase by the time of discharge, which was significant in patients with DF (p = 0.01). There was no difference in the leptin levels in patients with DF and DHF at any of the phases of illness and were similar to levels seen in healthy individuals (Fig 1C). Although levels of adiponectin were higher in patients who proceeded to develop DHF in the febrile phase compared to those with DF, this was not significant (Fig 1D). There were individual variations between the different adipokine levels in patients with DF and DHF between early illness and at the time of discharge, with some patients having their levels remaining the same, in some patients the levels decline while in some the levels become elevated (S1Fig).

In patients diagnosed with DF, a significant but weak correlation was observed between serum resistin and leptin levels during the febrile phase (r = 0.12, p = 0.01, Fig 2A), but not for patients with DHF (Fig 2B). Although serum resistin levels during the febrile phase significantly correlated with omentin levels of the febrile phase (Spearman's r = 0.35, p = 0.003) in all patients, this correlation was not seen when patients with DF (Fig 2C) and DHF (Fig 2D) were analysed separately. The overall omentin levels in the febrile phase also correlated with the AST (Spearman's r = 0.38, p = 0.001) and ALT levels (Spearman's r = 0.24, p = 0.04) as well as serum leptin levels with both AST (Spearman's r = 0.27, p = 0.02) and ALT (Spearman's r = 0.28, p = 0.02) in all patients. In separate analyses, we found no correlation between AST levels and serum resistin levels in patients with DF. However, in patients with DHF, during the febrile phase, there was a significant but weak correlation observed between serum resistin levels and AST levels (r = 0.21, p = 0.03, Fig 2F). In addition, a weak but significant correlation between serum resistin and ALT levels was seen in patients with DF (r = 0.08, p = 0.04, Fig 2G), but not in patients with DHF (Fig 2H). Additionally, a distinct inverse correlation was found between serum adiponectin and CRP levels during the febrile phase of patients diagnosed with DF (r = 0.11, p = 0.02, Fig 2I).

As adiponectin/leptin ratio has shown to be a reliable marker of adipose tissue dysfunction, we analysed the ratios of adiponectin in the febrile phase (adiponectin A) and the leptin levels in the febrile phase (leptin A), with clinical disease severity and laboratory parameters. Although there was no difference in adiponectin A/leptin A ratios in patients with DF and DHF (p = 0.26), this ratio inversely correlated with AST levels (Spearman's r = -0.26, p = 0.03) and ALT levels (Spearman's r = -0.34, p = 0.004).

9/71 patients had metabolic disease based on the presence of a history of diabetes, hypertension or hyperlipidaemia. Although serum omentin levels were higher in patients with metabolic disease (median of 899.9 pg/ml vs 578.7 pg/ml), this was not significant (p = 0.15).

## Relationship between viral loads, NS1 and adipokines

11(15.4%) of were infected with DENV1, 37 (52.1%) with DENV2 and 13 (18.3%) with DENV3. 46 (93.87%) with DF and 15 (68.18%) who progressed to develop DHF had detectable virus at the febrile phase. At the time of discharge, viraemia persisted in 22 (44.89%) in those with DF and 15 (68.18%) in those with DHF. Although not significant (p = 0.07), those who had DHF were more likely to have persistent viraemia at the time of discharge (odds ratio 2.6, 95% CI 0.88 to 6.9). However, at the time of presentation or discharge, there was no difference in the viral loads between patients who progressed to develop DHF and those who had DF.

As NS1 has shown to independently associate with clinical disease severity and disease pathogenesis [25, 26], we also assessed the relationship between NS1 and changes in adipokine levels. In the febrile phase NS1 antigen was positive in 30 (61.22%) with DF and 12 (54.54%) who proceeded to develop DHF. By the time of discharge, NS1 antigen was still positive in 17 (34.69%) in those with DF and 5 (22.72%) in those with DHF. The NS1 antigen was not more likely (p = 0.41) to be positive in those who had DHF (odds ratio 0.55, 95% CI 0.19 to 1.6). The serum leptin levels on admission significantly and inversely correlated with the viral loads at the time of discharge (Spearman's r = -0.31, p = 0.004). None of the other adipokines showed any relationship with the viral loads or with NS1 antigen levels.

## Relationship between serum IFNβ levels with adipokines, viral loads and inflammatory mediators

Although not significant (p = 0.14), serum IFNβ levels were lower in the febrile phase in those who progressed to develop DHF (median 0, IQR 0 to 39.4 pg/ml), compared to those who had

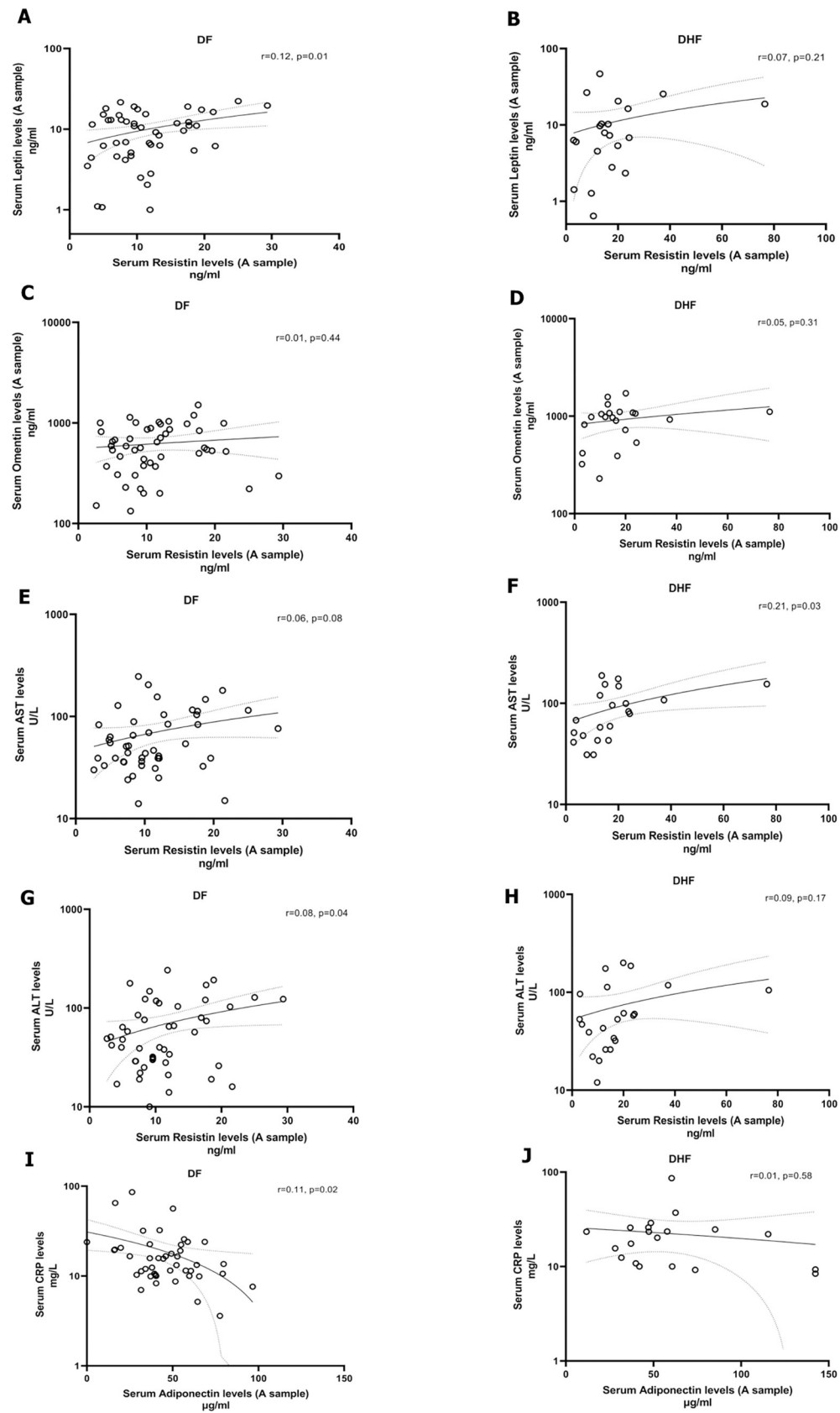

**Fig 2. Correlation adipokines with transaminase and CRP levels in patients with acute dengue.** Serum adipokine levels were measured by ELISA and liver transaminase levels and CRP levels were measured by automated biochemical analyzers during the febrile phase (n = 71). Among patients diagnosed with Dengue Fever (DF), a positive correlation was observed between serum resistin levels and leptin (Spearman's r = 0.12, p = 0.01) (A), whereas no significant correlation was found in patients with Dengue Hemorrhagic Fever (DHF) (Spearman's r = 0.07, p = 0.21) (B). Omentin levels during the febrile phase did not exhibit a significant correlation in either the DF (Spearman's r = 0.01, p = 0.44) (C) or the DHF groups (Spearman's r = 0.05, p = 0.31) (D). In the DHF group, AST levels showed a significant positive correlation with resistin levels during the febrile phase (Spearman's r = 0.21, p = 0.03) (F), whereas in the DF group, this correlation was not significant (Spearman's r = 0.06, p = 0.08) (E). Serum resistin levels were positively correlated with ALT levels in the DF group (Spearman's r = 0.08, p = 0.04) (G), but not in the DHF group (Spearman's r = 0.09, p = 0.17) (H). In the DF group, adiponectin exhibited a significant inverse correlation with CRP levels (Spearman's r = -0.11, p = 0.02) (I), whereas in the DHF group, no significant correlation was observed (Spearman's r = 0.01, p = 0.58) (J). The Spearman rank order correlation coefficient was used to evaluate the correlation between adipokine levels, liver transaminases and CRP levels.

DF (median 37.1, IQR 0 to 65.6 pg.ml) (Fig 3A). IFNβ levels inversely correlated with inflammatory markers such as CRP (Spearmans's r = -0.28, p = 0.03, Fig 3B) and with laboratory indicators of haemoconcentration such as the PCV (Spearman's r = -0.35, p = 0.01, Fig 3C). Although the CRP levels were higher in patients who progressed to DHF, this was not significant (p = 0.32). CRP levels significantly correlated with AST (p = 0.03) and ALT levels (p = 0.009). The adiponectin A: leptin A ratios also inversely correlated with the CRP levels (Spearman's r = -0.28, p = 0.01).

## Discussion

In this study we have shown that both resistin and omentin levels were significantly higher in the febrile phase in patients who progressed to develop DHF. Although leptin levels were not different at the febrile phase or discharge phase in patients with DF or DHF, leptin, resistin and omentin correlated with the AST and ALT levels, which are indicators of the extent of liver damage. We did not see any difference in the adipokine levels in those with metabolic disease compared to others, but the study was limited by the small number of patients with metabolic disease (n = 9). Therefore, it would be important to further investigate the role of these different adipokines in a larger patient cohort who have metabolic disease with acute dengue and the findings replicated.

Resistin is mainly produced by monocytes, macrophages and other leucocytes and its production is shown to be stimulated by lipopolysaccharide (LPS) and cytokines such as IL-6, TNFα and IL-1β, which are all shown to be elevated during early illness in dengue [27,28]. While the antiviral effects of resistin have not been studied extensively, it has been shown to induce IFNγ and also acts via TLR-4 to induce production of many proinflammatory cytokines and induce neutrophil extracellular trap formation [28]. We did find that the resistin levels were higher in the febrile phase in those who progressed to DHF, compared to those who had DF. However, it is not clear if resistin levels were induced by the high levels of proinflammatory cytokines which are known to be induced upon infection with the DENV and by dengue NS1[29], or if high resistin levels also contributed to the elevation of these cytokines.

Omentin has been considered to be an anti-inflammatory adipokine, which inversely correlated with waist circumference, dyslipidaemia, hypertension, impaired glucose tolerance and was shown to induce a protective effect against endothelial dysfunction in human umbilical vein endothelial cells [30]. However, in vitro studies have shown that omentin induces production of proinflammatory mediators from adipocytes and therefore is likely to contribute to obesity associated chronic inflammation [30]. In a study in obese individuals, it was shown that while higher serum levels of omentin-1 were associated with lower levels of IL-6 and TNFα, higher omentin was associated with IL-4, IL-13 and IL-1β, showing that different

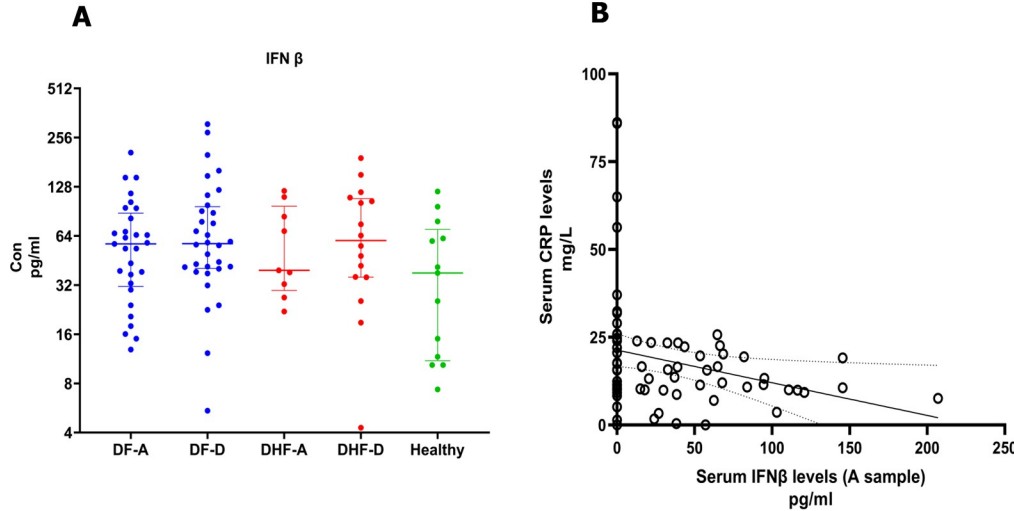

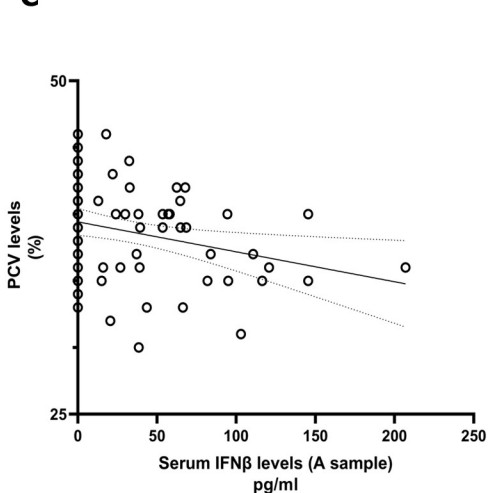

**Fig 3. Relationship between serum IFNβ levels with adipokines, viral loads and inflammatory mediators.** Serum IFNβ levels were measured in patients with DF (n = 49) and DHF (n = 22) during the febrile (A sample) and recovery phases (D sample), using a quantitative ELISA (A). The IFNβ levels inversely correlated with CRP (Spearmans's r = -0.28, p = 0.03) (B) and PCV (Spearman's r = -0.35, p = 0.01) (C). The Spearman rank order correlation coefficient was used to evaluate the correlation between IFNβ levels and inflammatory mediators. Wilcoxon matched pair singed-rank test was used to compare the levels of IFNβ of the febrile and the recovery phases of patients with DF and DHF. The Mann-Whitney U test (two tailed) was used to calculate the differences in the means in the DF and DHF cohorts. The error bars indicate the median and the interquartile ranges.

cohorts can show variable results regarding omentin levels [31,32]. In contrast, lower omentin levels were seen in patients with COVID-19 and omentin was thought to reduce IL-6 and other inflammatory cytokines in COVID-19[33]. In our study, we found that serum omentin levels were significantly higher in patients who progressed to develop DHF compared to DF and this difference was also seen at the time of discharge. Although the role of omentin in dengue is not known, omentin (intelectin) was shown to induce allergen induced production of IL-25, IL-33 and TSLP in asthma and atopic dermatitis [34]. We recently showed that innate

like lymphoid cells (ILC2s) were activated in dengue, were infected by DENV and showed an impaired type I interferon signature in those who had severe disease [18]. Therefore, it would be important to further study a potential role of omentin in inducing IL-25 and IL-33 which in turn activate ILC2 and thereby could be contributing to severe disease. Indeed, many studies have shown that asthma and allergy are independent risk factors for development of severe dengue [35].

Higher CRP levels during early illness in patients with dengue, was shown to associate with an increased risk of progression to severe disease and was also associated with hospitalization [36]. We found that CRP levels during early illness correlated with both AST and ALT levels, whereas serum adiponectin levels and adiponectin/leptin ratios during early illness inversely correlated with CRP levels. Adiponectin has shown to negatively correlate with highly sensitive CRP in patients with diabetes and was also shown to inversely correlate with CRP levels following the BNT162b2 vaccine in SARS-CoV-2 infected individuals [20,37]. As adiponectin inversely correlated with CRP and was correlated with the extent of the rise in liver transaminases, it is likely that this adipokine in likely to reduce inflammation and possibly immuno-pathogenesis in dengue, which should be further investigated.

In summary, in this study we found that adipokines such as resistin and omentin were significantly elevated during early illness in those who progressed to develop DHF. Many of these adipokines correlated with liver transaminases, while adiponectin inversely correlated with CRP levels. In order to identify therapeutic targets to prevent progression to severe dengue, it is crucial to further investigate the role of adipokines in disease pathogenesis. Specifically, future studies should focus on larger cohorts and explore the potential of adipokines as prognostic markers for predicting disease severity and outcomes in dengue patients. By conducting such research, we can gain a better understanding of the pathogenesis of severe dengue and potentially identify novel pathways for treatment and prevention.

## Supporting information

**S1 Fig. Adipokine levels in patients with acute dengue.** Resistin, omentin, leptin, and adiponectin levels were measured in patients with DF (n = 49) and DHF (n = 22) during the febrile and recovery phases, using a quantitative ELISA. The Wilcoxon matched pair singed-rank test was used to compare the levels of adipokines of the febrile (A sample) and the recovery phases (D sample) of patients with DF and DHF. The Mann-Whitney U test (two tailed) was used to calculate the differences in the means in the DF and DHF cohorts. The error bars indicate the median and the interquartile ranges.
(TIF)

**S1 Data. Raw data of the clinical disease severity, viral loads, adipokine levels and cytokines of patients used in the study.**
(XLSX)

## Author Contributions

**Conceptualization:** Graham S. Ogg, Gathsaurie Neelika Malavige.

**Data curation:** Heshan Kuruppu, W. P Rivindu H. Wickramanayake, Deneshan Peranantharajah, H. S. Colambage.

**Formal analysis:** Heshan Kuruppu, Gathsaurie Neelika Malavige.

**Funding acquisition:** Chandima Jeewandara, Graham S. Ogg, Gathsaurie Neelika Malavige.

**Investigation:** Heshan Kuruppu, W. P Rivindu H. Wickramanayake, Lahiru Perera, Laksiri Gomes.

**Project administration:** Chandima Jeewandara, Ananda Wijewickrama, Gathsaurie Neelika Malavige.

**Resources:** Chandima Jeewandara, Ananda Wijewickrama, Graham S. Ogg.

**Supervision:** Chandima Jeewandara, Gathsaurie Neelika Malavige.

**Writing – original draft:** Heshan Kuruppu, Gathsaurie Neelika Malavige.

**Writing – review & editing:** Graham S. Ogg, Gathsaurie Neelika Malavige.

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
