## [Decision Letter · Decision Letter 0]

24 Mar 2023

Dear Professor Malavige,

Thank you very much for submitting your manuscript "Adipokine levels and their association with clinical disease severity in patients with dengue" for consideration at PLOS Neglected Tropical Diseases. As with all papers reviewed by the journal, your manuscript was reviewed by members of the editorial board and by several independent reviewers. In light of the reviews (below this email), we would like to invite the resubmission of a significantly-revised version that takes into account the reviewers' comments. 

The reviewers see the value of the study, but there are significant concerns about the study limitations (i.e., relative small size) that must be addressed in the discussion (with justification of the statistics used), as well as the absence of plasma lipid biomarker data. Please pay particular attention to the additional assays suggested by reviewer 2.

We cannot make any decision about publication until we have seen the revised manuscript and your response to the reviewers' comments. Your revised manuscript is also likely to be sent to reviewers for further evaluation.

Sincerely,

Benjamin L. Makepeace

Academic Editor

Andrea Marzi

Section Editor

The reviewers see the value of the study, but there are significant concerns about the study limitations (i.e., relative small size) that must be addressed in the discussion (with justification of the statistics used), as well as the absence of plasma lipid biomarker data. Please pay particular attention to the additional assays suggested by reviewer 2.

Reviewer's Responses to Questions

**Key Review Criteria Required for Acceptance?**

**Methods**

-Are the objectives of the study clearly articulated with a clear testable hypothesis stated?

-Is the study design appropriate to address the stated objectives?

-Is the population clearly described and appropriate for the hypothesis being tested?

-Is the sample size sufficient to ensure adequate power to address the hypothesis being tested?

-Were correct statistical analysis used to support conclusions?

-Are there concerns about ethical or regulatory requirements being met?

Reviewer #1: Kuruppu et al have measured serum adipokines, a group of hormones released by fat cells, in patients early on in the clinical course of dengue infection, and then later in recovery. There has been very little work in this area and I believe this is the first report of this kind. A test to predict who will go on to develop severe disease would be very welcome, and although this paper mainly addresses adults these are still potentially helpful data, with some caveats.

The work is high quality and is written up well. The main deficiency is in the relatively small sample size. With some modifications, the work can be recommended for publication.

General points:

One issue which is common in papers like this is there is no attempt at a sample size calculation, nor a post hoc attempt to describe what sort of difference could have been reliably detected using the number of patients with samples available. Sometimes it can be challenging in these settings to recruit adequate patient numbers, not having a pre-specified target does not invalidate the work. Nevertheless it would be helpful for the reader to be able to assess how much could be due to chance, and if they were to repeat this work, what sort of sample they should be aiming for.

In terms of the specific review points, some general revisions such as stating the hypothesis tested more clearly, etc, could be better.

Reviewer #2: The work by Kuruppu et al. aims to explore the association between adipokine expression and dengue severity. The team enrolled 71 subjects, 49 with diagnosed with dengue fever and 22 with dengue hemorrhagic fever. A number if adipokines (resistin, omentin, leptin, and adiponectin) are measured in patients at admission and discharge. The research team also measured key liver enzymes, viral load, NS1 and the anti-inflammatory cytokine, IFNb. While the team has generated a nice dataset there are a few limitations.

1. Given the major focus is on adipokines, the paper would benefit from measuring plasma lipid biomarkers. This is especially relevant since the Sri Lankan diet consists of saturated oils such as coconut oil and therefore preexisting lipid levels may confound the interpretation of the adipokine expression data. If sample does not exist to do anymore plasma analysis, could the BMI of each patient be calculated? The BMI may act as a surrogate for lipid profile. 

2. The study team should consider providing more demographic information of the dengue cohort. For example, the paper would benefit from a table that has not only age, gender (which is noted in the manuscript as text) but also comorbidities such as percent with diabetes, cardiovascular disease, COVID-19 etc. This information can then be used to reinterpret data from Figure 1.

3. There is little to no information about the healthy cohort. Are these individuals age-matched, gender-matched, diet-matched or clinically-matched for other underlying ailments? This is relevant given that adipokine levels can change as a result of all these variables. In addition, the median of the healthy group look similar to the infected groups at admission, which suggest that what the authors are seeing maybe due to other factors. The only adipokine that demonstrates marked differences between the healthy and infected groups is omentin.

4. The Spearman correlations albeit significant are quite weak with a Spearman's coefficient which ranges from 0.3-0.4. Therefore it should be noted in the manuscript that these are fairly weak.

Reviewer #3: -Can the authors describe the clinical features suggestive of acute dengue that were the inclusion criteria for recruitment?

-What are the exclusion criteria?

-Please mention, the total number of patients that were recruited based on the clinical features, prior to the inclusion of 71 patients for the study.

-It is not clear whether this was a retrospective/prospective study or did the authors work on archived blood specimens from an earlier study.

-Was there enough statistical power to test the study's hypotheses?

-Please describe the study period/patient recruitment period?

-Can the authors include the number of days since disease onset before the DF patients progressed to develop DHF.

-Blood specimen collection from healthy individuals for measurement of adipokines and INF-beta levels must be described.

**Results**

-Does the analysis presented match the analysis plan?

-Are the results clearly and completely presented?

-Are the figures (Tables, Images) of sufficient quality for clarity?

Reviewer #1: see below

Reviewer #2: 1. The authors should consider adding more information to Table 1 such as age, gender, and prior clinical histories.

2. The authors should add a column for the p-values comparing the adipokine levels at admission and discharge. This makes it much easier for the reader.

3. The authors may want to consider a different way to visualize the data in Figure 1. For example rather than dot plots maybe spaghetti plots to see the change of each cytokine for a given individual between admission and discharge.

4. The authors should also consider presenting Figure 2 for the DF and DHF groups separately.

5. The authors should consider presenting a figure of the adipokine levels for those patients from the dengue fever group who progressed on to get DHF. Is there a difference in particular adipokines? Given that data is presented for the entire cohort it would be important to see if those patients that went on to get DHF expressed more of one marker over the other.

6. The paper would have benefited from an analysis of different cellular subsets. In the cohort, how dysregulated was the T cell repertoire? Given that M1 macrophages are induced by adipokines, did the cohort express these cells. If the study team has access to the cells I would strongly suggest this piece of work.

Reviewer #3: -Table 1, the abbreviations (eg. IQR, A, D) should be described.

-Table 1, the empty boxes should be labelled 'not applicable' or 'not performed'.

-The viral loads seemed low to me. It is a little odd to find acute viral loads at these levels. 

-Is there any explanation why leptin levels are relatively similar between Healthy, DF and DHF?

**Conclusions**

-Are the conclusions supported by the data presented?

-Are the limitations of analysis clearly described?

-Do the authors discuss how these data can be helpful to advance our understanding of the topic under study?

-Is public health relevance addressed?

Reviewer #1: Line 335 – “adipokine IS likely to reduce” Also this statement is overstated – there was no difference in adiponectin levels with disease severity. The relationship is only with another marker. The clinical relevance of this is still to be shown – a larger study may be needed. Suggest this is revised to be more circumspect and to leave this as a hypothesis for future study.

The same is true of IFN beta in the discussion, as above for adipokines. This should be more cautiously worded. Having a better idea of the power of the study would also help here, including with corrections for multiple measurements (I appreciate this may not be straightforward due to the co-linear nature of some of the variables).

There are many interesting points in this article, which should be published. But the above points should be revised first.

Reviewer #2: The paper by Kuruppu et al. provides a descriptive analysis of the association between adipokines and dengue severity. However, the authors may be overstating some of their conclusions given the weak correlations seen in Figure 2. Therefore, the authors should consider tempering their conclusions to be more suggestive. Furthermore, the authors should explain the rationale to measure IFNbeta and to use NS1. The study team measures NS1, provides the results but there is no mention in the conclusions. If NS1 was used to differentiate between the different dengue serotypes, this should be mentioned.

The topic is interesting and worth publishing given the importance of understanding how dengue immunity can be shaped by factors such as diet, obesity etc. The research team performs a nice piece of work but several modifications are required before publication.

Reviewer #3: The authors showed the potential of adipokines for use as prognostic markers for dengue. However, it must be noted that this study was performed on a comparatively low number of study participants. The present study is best described as a preliminary study showing a variety of adipokines with different expression levels between patients with DF and DHF. A larger cohort is required to corroborate the findings from this study.

**Editorial and Data Presentation Modifications?**

Reviewer #1: More specific points:

Line 75-76 – whilst the point about co-morbidities is well made, it would be more generalisable to say that severe disease is commonly seen in children, who usually don’t have such co-morbidities, but that these co-morbidities are highly relevant in adults, who can still develop severe disease.

Line 79 – “infected with DENV recover” (no need for “the”, and recover not recovery)

Line 83 – 91 Re COVID-19 and influenza: - true, but not really relevant or needed (eg also true in HIV, TB, sepsis, and many others etc etc). Better to point out that these co-morbidities are associated with worse outcomes from many infections as well as dengue. this isn’t a controversial statement and shouldn’t really need discussion of specific diseases to back it up. I see that later some of the inspiration comes from the COVID-19 literature – these points are well made but the way the comparison is introduced feels somewhat artificial.

Line 134 – “fluid leakage WAS assessed”

Line 151 DEN or DENV?

Line 172/3 – I don’t believe that space is particularly limited in PLOS NTD – I would strongly suggest that suppl table 1 is moved into the main text. This would be important information and with PLOS journals the supplements are always individual files, each of which have to downloaded – which is a pain. (Take note PLOS editorial teams!!)

Line 199-200 – how does recovery get called “D??” Why not just call this acute and recovery, or A and Rec if you really need it to be short? (I think you could fit in DHF with acute or recovery below.)

What correction for multiple comparisons has been conducted for Fig 1?

There’s one outlier in Fig 2 for resistin levels – what happens when you take this sample out? Does any of the effect depend on this one sample? It would be much clearer if the r and p values from the Spearman tests were indicated on each panel in Fig 2 – please annotate them with this. (And indeed for all figs with this analysis.)

Line 224 – “As adiponectin/leptin ratio has BEEN shown to be a reliable marker of adipose tissue dysfunction” – Been missing. Also this needs a reference – it is I presume not referring to the present work.

Line 321 – “infected by DENV” (no “the”)

Reviewer #2: page 4, line 79 - remove "the" and change "recovery" to "recover"

page 4, line 81 - consider rephrasing this sentence to "those who develop DHF have an altered and dysfunctional immune response characterized by ..."

page 4, line 85 - remove "as seen with" and replace with "and"

Reviewer #3: (No Response)

**Summary and General Comments**

Reviewer #1: see abo e

Reviewer #2: This manuscript would benefit from the following experiments:

1) Measure plasma lipid markers

2) If cells are available for the different time points, FACS assays to explore T cell subsets and M1 macrophages would be enlightening.

3) The omentin data is intriguing given that this adipokine is the only one that was higher in the dengue-infected groups as compared to the healthy group. The team should consider measuring IL-25 in the study groups.

4) The authors should consider re-analyzing the data by clinical history to assess whether new observations can be made. In examining the cohort as a bulk group the authors may be missing out on some interesting data.

Reviewer #3: This is an interesting study looking at the different adipokine levels between DF and DHF patients. It would be even interesting to see the different adipokine levels between patients infected with different dengue serotypes. My major concern is on the number of study participants, which in my opinion, has to be larger; especially in Sri Lanka where dengue is endemic. There are several typos too (Line 79-recovery, Table 1-0.34 to 0.07 to 4 & 0.3 to 0.17 to 1).

PLOS authors have the option to publish the peer review history of their article (what does this mean?). If published, this will include your full peer review and any attached files.

Reviewer #1: No

Reviewer #2: No

Reviewer #3: No
---

## [Decision Letter · Decision Letter 1]

17 Jul 2023

Dear Professor Malavige,

Thank you very much for submitting your manuscript "Adipokine levels and their association with clinical disease severity in patients with dengue" for consideration at PLOS Neglected Tropical Diseases. As with all papers reviewed by the journal, your manuscript was reviewed by members of the editorial board and by several independent reviewers. The reviewers appreciated the attention to an important topic. Based on the reviews, we are likely to accept this manuscript for publication, providing that you modify the manuscript according to the review recommendations. 

Reviewer 1 retains some significant concerns about your manuscript, specifically the thoroughness of the revision with regard to statistical power of the study, and the limitations that must be expressly dealt with in the Discussion. Please address these points in sufficient detail.

Sincerely,

Benjamin L. Makepeace

Academic Editor

Andrea Marzi

Section Editor

Reviewer 1 retains some significant concerns about your manuscript, specifically the thoroughness of the revision with regard to statistical power of the study, and the limitations that must be expressly dealt with in the Discussion. Please address these points in sufficient detail.

Reviewer's Responses to Questions

**Key Review Criteria Required for Acceptance?**

**Methods**

-Are the objectives of the study clearly articulated with a clear testable hypothesis stated?

-Is the study design appropriate to address the stated objectives?

-Is the population clearly described and appropriate for the hypothesis being tested?

-Is the sample size sufficient to ensure adequate power to address the hypothesis being tested?

-Were correct statistical analysis used to support conclusions?

-Are there concerns about ethical or regulatory requirements being met?

Reviewer #1: See below

Reviewer #3: The authors have revised the manuscript accordingly and in its current form, the manuscript contains additional information to help readers appreciate the work that was performed.

**Results**

-Does the analysis presented match the analysis plan?

-Are the results clearly and completely presented?

-Are the figures (Tables, Images) of sufficient quality for clarity?

Reviewer #1: See below

Reviewer #3: The authors have improved the results section (Table 1) accordingly.

**Conclusions**

-Are the conclusions supported by the data presented?

-Are the limitations of analysis clearly described?

-Do the authors discuss how these data can be helpful to advance our understanding of the topic under study?

-Is public health relevance addressed?

Reviewer #1: See below

Reviewer #3: Due to the relatively small sample size in the present work, the authors rightly suggested and concluded that adipokines are likely to play some role in dengue pathogenesis and this could be a basis for further work on larger cohorts and to investigate the potential of adipokines as prognostic markers in predicting disease severity and outcomes in

dengue patients.

**Editorial and Data Presentation Modifications?**

Reviewer #1: See below

**Summary and General Comments**

Reviewer #1: I’m grateful to the authors for their time in responding to my points about the manuscript. I remain of the view that these data are helpful to have in the published literature. Unfortunately, I don’t think the authors have gone far enough yet in addressing the limitations of their study.

Overall I find the response regarding the statistics to be unsatisfactory. There is not sufficient discussion about the limitations of the study in the discussion. There is a lot of discussion about the possible biology of adipokines in inflammation. But this discussion may not be so relevant if the findings cannot be replicated. I can’t quite follow the sample size calculation – to detect a 50% change in a marker can be done with far fewer people depending on what the variance of the data is, I don’t see this clearly addressed. The following need to be addressed in the manuscript for it to be recommended for publication:

1. sample size and maximum difference that could have been detected

2. specify whether multiple comparisons have been conducted or not

3. Section on limitations of the study which should specifically acknowledge these weaknesses and point out that these findings need to be replicated

4. removal of most of the discussion about the biology of the markers, eg especially the comments about adipokines and IFN beta in the discussion (line 335 and below in the original manuscript) remain overstated – on further reflection I think these should be removed altogether.

Much of the discussion could be replaced with a shorter paragraph pointing out there is biological plausibility for these markers to be relevant, with expansion of the discussion of limitations and the preliminary nature of these data.

If any subsequent response could have line numbers in the modified document so it is easier to see the changes that would be helpful.

Making the raw data available will help anyone interested in doing similar work.

There are some additional specific points below:

Regarding point 7 – if D indicates discharge, the please use a clearer abbreviation such as “Disch.”

“As the reviewer has suggested, we have included a more details description of the hypothesis in the revised version of the manuscript.” – please indicate where this is – I do not see this to my satisfaction! If this was exploratory, then it would better to say so. It does not detract from the work to state this.

“between a defunct type I interferon response” – defective type I interferon

Please relabel the axes of Fig 1 so that the reader can appreciate this more clearly – DF-A should be “Acute” and DF-D “Disch” – I suggest then putting the disease state underneath as a sub title. This will be a lot clearer.

“Unlike as seen with the virus, the NS1 antigen was not more likely (p=0.41) to be positive in those who had DHF (odds ratio 0.55, 95% CI 0.19 to 1.6).” – please remove “unlike as seen with the virus – as this was not significant either.

“However, when analyzed separately, while there was no correlation of AST levels with serum resistin levels in patients with DF, serum resistin levels during the febrile phase were significantly correlated with AST levels (r=0.21, p=0.03, Figure 2F) in those with DHF, although this association was weak.” – suggest splitting this into two sentences

“possible due to small number (n=9) of individuals” – suggest replace with “but the study was limited by the small number of patients with metabolic disease (n=9).”

Reviewer #3: (No Response)

PLOS authors have the option to publish the peer review history of their article (what does this mean?). If published, this will include your full peer review and any attached files.

Reviewer #1: No

Reviewer #3: No

Figure Files:

Data Requirements:

Reproducibility:

References

---

## [Editor Report · Decision Letter 2]

4 Aug 2023

Dear Professor Malavige,

Thank you very much for submitting your manuscript "Adipokine levels and their association with clinical disease severity in patients with dengue" for consideration at PLOS Neglected Tropical Diseases. As with all papers reviewed by the journal, your manuscript was reviewed by members of the editorial board and by several independent reviewers. The reviewers appreciated the attention to an important topic. Based on the reviews, we are likely to accept this manuscript for publication, providing that you modify the manuscript according to the review recommendations. 

From a scientific perspective, I am satisfied that you have revised the manuscript to the sufficient standard. However, note that it is a condition of publishing in PLoS journals that all non-sensitive raw data are made publicly available. For more details, please see https://journals.plos.org/plosntds/s/data-availability

Sincerely,

Benjamin L. Makepeace

Academic Editor

Andrea Marzi

Section Editor

From a scientific perspective, I am satisfied that you have revised the manuscript to the sufficient standard. However, note that it is a condition of publishing in PLoS journals that all non-sensitive raw data are made publicly available. For more details, please see https://journals.plos.org/plosntds/s/data-availability

Figure Files:

Data Requirements:

Reproducibility:

References

---

## [Editor Report · Decision Letter 3]

21 Aug 2023

Dear Professor Malavige,

We are pleased to inform you that your manuscript 'Adipokine levels and their association with clinical disease severity in patients with dengue' has been provisionally accepted for publication in PLOS Neglected Tropical Diseases.

Best regards,

Benjamin L. Makepeace

Academic Editor

Andrea Marzi

Section Editor

---

## [Editor Report · Acceptance letter]

5 Sep 2023

Dear Professor Malavige,

We are delighted to inform you that your manuscript, "Adipokine levels and their association with clinical disease severity in patients with dengue," has been formally accepted for publication in PLOS Neglected Tropical Diseases.

Best regards,

Shaden Kamhawi

co-Editor-in-Chief

Paul Brindley

co-Editor-in-Chief
